# Sex Differences in Swimming Disciplines—Can Women Outperform Men in Swimming?

**DOI:** 10.3390/ijerph17103651

**Published:** 2020-05-22

**Authors:** Beat Knechtle, Athanasios A. Dalamitros, Tiago M. Barbosa, Caio Victor Sousa, Thomas Rosemann, Pantelis Theo Nikolaidis

**Affiliations:** 1Medbase St. Gallen Am Vadianplatz, 9001 St. Gallen, Switzerland; 2Institute of Primary Care, University of Zurich, 8091 Zurich, Switzerland; thomas.rosemann@usz.ch; 3Faculty of Physical Education and Sport Sciences, School of Physical Education and Sports Science, Aristotle University of Thessaloniki, 55535 Thessaloniki, Greece; dalammi9@hotmail.com; 4Physical Education and Sport Science Academic Group, National Institute of Education, Nanyang Technological University, Singapore 637616, Singapore; tiago.barbosa@nie.edu.sg; 5Research Centre in Sports, Health and Human Development, 5001-801 Vila Real, Portugal; 6Department of Sport Sciences, Polytechnic Institute of Bragança, 5300-253 Bragança, Portugal; 7Bouve College of Health Sciences, Northeastern University, Boston, MA 02115, USA; cvsousa89@gmail.com; 8Exercise Physiology Laboratory, 18450 Nikaia, Greece; pademil@hotmail.com; 9School of Health and Caring Sciences, University of West Attica, 12243 Athens, Greece

**Keywords:** gender difference, sex gap, swimming performance, swimming stroke, holistic approach

## Abstract

In recent years, the interest of female dominance in long-distance swimming has grown where several newspaper articles have been published speculating about female performance and dominance—especially in open-water ultra-distance swimming. The aim of this narrative review is to review the scientific literature regarding the difference between the sexes for all swimming strokes (i.e., butterfly, backstroke, breaststroke, freestyle and individual medley), different distances (i.e., from sprint to ultra-distances), extreme conditions (i.e., cold water), different ages and swimming integrated in multi-sports disciplines, such as triathlon, in various age groups and over calendar years. The influence of various physiological, psychological, anthropometrical and biomechanical aspects to potentially explain the female dominance was also discussed. The data bases Scopus and PUBMED were searched by April 2020 for the terms ’sex–difference–swimming’. Long-distance open-water swimmers and pool swimmers of different ages and performance levels were mainly investigated. In open-water long-distance swimming events of the ’Triple Crown of Open Water Swimming’ with the ’Catalina Channel Swim’, the ’English Channel Swim’ and the ’Manhattan Island Marathon Swim’, women were about 0.06 km/h faster than men. In master swimmers (i.e., age groups 25–29 to 90–94 years) competing in the FINA (Fédération Internationale de Natation) World Championships in pool swimming in freestyle, backstroke, butterfly, breaststroke, individual medley and in 3000-m open-water swimming, women master swimmers appeared able to achieve similar performances as men in the oldest age groups (i.e., older than 75–80 years). In boys and girls aged 5–18 years—and listed in the all-time top 100 U.S. freestyle swimming performances from 50 m to 1500 m—the five fastest girls were faster than the five fastest boys until the age of ~10 years. After the age of 10 years, and until the age of 17 years, however, boys were increasingly faster than girls. Therefore, women tended to decrease the existing sex differences in specific age groups (i.e., younger than 10 years and older than 75–80 years) and swimming strokes in pool-swimming or even to overperform men in long-distance open-water swimming (distance of ~30 km), especially under extreme weather conditions (water colder than ~20 °C). Two main variables may explain why women can swim faster than men in open-water swimming events: (i) the long distance of around 30 km, (ii) and water colder than ~20 °C. Future studies may investigate more detailed (e.g., anthropometry) the very young (<10 years) and very old (>75–80 years) age groups in swimming

## 1. Introduction

Swimming is a specific sports discipline which can be performed in a range of styles, usually referred to as ’strokes’ [1,2,3,4], over different lengths [5,6] and in both pools (i.e., indoor, outdoor) of different lengths (mainly 25 m and 50 m) and in open water (i.e., sea, lake, river) [7,8,9,10]. The different swimming strokes are butterfly [1], backstroke [2], breaststroke [3], freestyle [4] and individual medley referred to the combination of the four different strokes [11]. In addition to these individual events, four swimmers can take part in either a freestyle or medley relay. In pool-swimming competitions, the distances for butterfly, backstroke, breaststroke and freestyle usually include 50 m to 200 m, whereas individual medley is held over 200 m and 400 m [11]. In freestyle, the 800 m and the 1500 m are further race distances in pool-swimming [4,12]. Indoor-swimming events with a defined time limit (i.e., 12 h) are also held [13]. In open-water swimming, master swimmers most often compete in 3000 m [14], whereas 5 km [6,10], 10 km [10,15] and 25 km [10,15] races were held for elite swimmers. Open-water swimming events of different lengths in lakes and seas are held as solo swims [7]. Swimming is also part of multi-sports races like triathlons over different distances like the Olympic Distance triathlon [16,17], the half-Ironman [18], the Ironman [16,19] and longer triathlon race distances than the Ironman distance [20].

In recent years, the interest of female dominance in long-distance swimming grew where several newspaper articles were published speculating about the female performance and dominance especially in open-water ultra-distance swimming. In one newspaper article, the history of female performance in open-water swimming started with Gertrud Ederle in the ’English Channel Crossing’ and the female dominance in ’Manhattan Island Marathon Swim’. The author discussed the problem of comparing the fastest men and women in contrast to all men and women, leading to a different finding regarding male or female dominance. Particularly, the fastest men beat the fastest women, but that the average woman was faster than the average man [21]. In addition, another newspaper article describing the sex difference in swimming and running discussed the aspect of the fastest women and men [22]. A further newspaper article cites Steven Munatones, one of the world’s top experts on open-water swimming, reporting that the average female time was 33 min faster than the average male time in the 135 years of the ’English Channel Crossing’ [23]. It seemed that women were better in long-distance open-water swimming. A further newspaper article reported that the female swimmer Sarah Thomas was the first person in the world to cross the ’English Channel’ four times in a row without stopping [24]. In a further newspaper article, women were described winning also ultra-endurance races in cycling and running ahead of all men [25]. Moreover, a newspaper article reported that the fastest women ever were faster than the fastest men ever in both the ‘Catalina Channel Swim’ and in the ’Manhattan Island Marathon Swim’ held in the USA [26].

These descriptions lead to the intention to review existing literature to confirm or disprove these statements of female dominance in open-water swimming. Since sex-related differences including anthropometric characteristics, swimming energy, as well as stroking parameters have been previously reported [27], it would also be interesting to examine this ’sex gap’ as translated into swimming performance. Such information would have both theoretical and practical relevance. From a theoretical point of view, the sex difference in human performance has been a major topic in exercise physiology, and thus, researchers working in this area would benefit from new knowledge on sex differences in swimming. From a practical perspective, coaches usually working with both sexes could use sex difference in swimming to optimize the training of their athletes. Therefore, the aim of the present research was to review original studies on sex differences in swimming performance with regards to age, swimming strokes and race distance.

## 2. Method

The data bases Scopus and PUBMED were searched on April 2020 for the terms ’sex–difference—swimming’. The search in Scopus led to 239 entries, the search in PUBMED to 558 entries. We excluded case studies, case reports, animal studies, studies with divers and rowers and studies with patients. Although the newspaper articles primarily reported about outstanding female achievements in open-water long-distance swimming, we consider in this narrative review all scientific results upon differences between the sexes in swimming for all swimming strokes (i.e., butterfly, backstroke, breaststroke, freestyle and individual medley), distances (i.e., from sprint to ultra-distances), conditions (i.e., cold water), ages (i.e., youth and master swimmers) and swimming integrated in multi-sports disciplines such as triathlon.

## 3. Findings

### 3.1. Pool-Swimming

Swimming competitions are held in pool-swimming in short-course (i.e., 25 m or 25 yards) and long-course (i.e., 50 m) pools from 50 m to 200 m in the four different strokes such as butterfly, backstroke, breaststroke and freestyle [28,29], where freestyle races were also held for 400 m, 800 m and 1500 m [29]. Apart from distance-limited swimming races, also time-limited swimming events (i.e., 12 h) are performed in pool-swimming [13]. Studies investigated different populations such as elite swimmers competing at national and international level for different strokes and distances, youth and master swimmers [1,2,3,8,29,30,31]. In pool-swimming, it seemed that the sex difference varied with the distance of the events [29,30,32]. For elite swimmers competing in different strokes such as freestyle [28], butterfly [31], breaststroke [30] and individual medley [30], the sex difference decreased with increasing race distance. In long-distance pool-swimming such as a 12-h-swim, women were able to achieve a similar performance to men. In the ’Zurich 12-h Swim’ held in Switzerland, the annual best performance did not differ between males (~38.3 km) and females (~34.4 km), respectively [8]. For master swimmers competing at the FINA World Championships in different age groups for different strokes and distances, men were faster than women for all strokes, distances and age groups except in 50–800-m freestyle for age groups 80–84 and 85–89 years [4], in 50–200-m butterfly for age group 90–94 years [1], in 200-m and 400-m individual medley for age groups 85–89 und 90–94 years [11], in 50–200-m breaststroke for age groups 90–94 and 95–99 years [3] and in 50–200-m backstroke for age groups 85–89, 90–94 and 95–99 years [2] where women achieved a similar performance than men. The disparate findings were explained by differences in performance level, race distance, stroke, age and sample size.

### 3.2. Open-Water Swimming

In addition to pool-swimming events, open-water swims were held as individual swims (i.e., solo swims in Channel Crossings without drafting) or competitions in long-distance swimming events up to 25 km in open-water swimming [10,15] where swimmers are allowed to swim in a group and where drafting is allowed. In addition, there were open-water swimming events of different lengths in lakes and seas [7] where drafting is also allowed.

In open-water swimming, the water temperature may be of importance for female dominance. For official swimming competitions held in heated pools, the FINA has established that water temperature shall be at 25 °C to 28 °C [33]. Apart from swimming competitions held in heated pools, swimming events were also held in open water such as rivers, lakes and seas. For open-water swimming events sanctioned by the FINA (i.e., World Cup races in 5 km, 10 km and 25 km), the FINA has established that the water temperature should be a minimum of 16 °C and a maximum of 31 °C [34]. In Channel Crossings like the ’English Channel Swim’, swimmers face, however, a water temperature of 15 °C at the end of June, increasing to 18 °C by the beginning of September [35]. Water temperature seemed to have an influence on the performance of the swimmers. For open-water swimmers competing in the ‘Marathon Swim in Lake Zurich’, a 26.4 km open-water ultra-swim held in Switzerland, performance of the top swimmers was negatively related to water temperature [9].

Female performance was investigated for different solo swims where drafting is not possible. In these studies, different groups of different performance levels were investigated. It seemed that women were able to achieve a similar performance to men in solo swims in long-distance open-water swimming such as individual Channel Crossings [7,36,37], where water temperatures were generally below 20 °C. In lake swimming such as ‘Marathon Swim Lake Zurich’ with water temperatures at or warmer than 20 °C, women achieved a similar performance to men [9].

However, depending upon the investigated sample (i.e., the fastest woman/man, the three fastest women/men, the five fastest women/men, the ten fastest women/men, all women/men, the annual fastest women/men, the annual three fastest women/men), women were able to outperform men in long-distance open-water swimming [36,37,38]. In the ‘English Channel Crossing’, the overall female swim time of 13:16 h:min was not different compared to the overall male swim time of 13:35 h:min between 1875 and 2011 [7]. Although the fastest male swim time (6:57 h:min) during this period was 6.7% faster than the fastest female swim time (7:25 h:min), the sex difference in performance of the top three times was ~8.9% [7]. The fastest annual swim speed did not differ between men (~0.89 m/s) and women (~0.84 m/s) [8]. In the ’Triple Crown of Open Water Swimming’ with ’Catalina Channel Swim’, ’English Channel Swim’ and ’Manhattan Island Marathon Swim’, overall women were ~0.06 km/h faster than overall men [31]. However, women were ~0.07 km/h slower than men when considering the annual five fastest swimmers [37]. Analyses were also performed for the single events of the ’Triple Crown of Open Water Swimming’. In the ’Catalina Channel Swim’, the fastest woman ever was faster than the fastest man ever (~22 min) [36]. The three fastest women were faster than the three fastest men (~20 min), however, the difference reached no statistical significance [36]. The ten fastest women were ~1 min faster than the ten fastest men, however, also here, the difference reached no statistical significance [36]. However, the annual fastest women (~10:51 h:min) were ~52.9 min (~16%) faster than the annual fastest men [36]. In a further open-water event held in the USA, women were faster than men. In the ’Manhattan Island Marathon Swim’, the ten fastest women were ~12%–14% faster than the ten fastest men [38]. Open-water swimming is also held for master swimmers competing in 3000 m. In master swimmers competing at the FINA World Championships, men were faster than women for all age groups except age groups 75–79, 80–84 and 85–89 years where women achieved the same performance like men [14]. Not only the age, but also the distance may be of importance. It seemed that in shorter open-water swimming events with a higher water temperature, women have a disadvantage compared to men. In the ’Marathon Swim in Lake Zurich’, the male record was 2.3% faster than the female record. For the annual winners, men were ~11.5% faster [9].

Based on these observations, in most cases analyzed, there were no sex differences in performance during open-water swimming. In some instances, women were faster than men. However, a variety of parameters such as water temperature and distance can influence the outcome. Depending upon the sample size (i.e., the fastest woman/man, the three fastest women/men, the five fastest women/men, the ten fastest women/men, all women/men, the annual fastest women/men, the annual three fastest women/men) and the statistical approach (i.e., comparison of groups or comparison of changes over time), women were faster than men in this specific sports discipline.

### 3.3. Ice Swimming

Since 2009, ice swimming for 1 mile and 1 km is a new discipline in open-water swimming [39,40]. In this swimming discipline, water temperature must be colder than +5 °C. One may assume that female performance may be better when water temperature is very low. Performances of women and men were investigated for ’Ice Mile’ and ’1 km Ice event’ where the fastest men were faster than the fastest women in both events [41]. Obviously, women had no advantage in this cold water. In the ’Ice Mile’, variables such as calendar year, number of swims, water temperature and wind chill showed no relation to swimming speed for both women and men. Water temperature was not correlated to swimming speed in either ’Ice Mile’ or ’1 km Ice event’ for both women and men [41]. Therefore, the limited data concerning ice swimming demonstrate that men have an advantage, compared to women, in this specific condition. Regarding the results from long-distance open-water swimming, the length of the event may be decisive. Future studies may investigate the body composition of open-water long-distance swimmers and ice swimmers. There may be a difference in body fat in the competitors in the two disciplines.

### 3.4. Age

Age is an important aspect regarding the sex difference in swimming performance. This aspect was investigated for different age groups, strokes and swimming distances. An analysis of sex differences in swimming speed for the top-10 World ranking (i.e.,1st–10th place), age group (25–89 years), and event distance from the world’s top ten swimming times of both women and men in the World Championships showed that the sex difference in swimming speed increased with world record place and age [42]. Very recent studies investigated the performance trends and sex difference in swimming performance in master swimmers competing in the FINA World Championships in pool-swimming in freestyle [4], in backstroke [2], in butterfly [1], in breaststroke [3], in individual medley [11] in 3000-m open-water swimming [14] and for youth swimmers [43]. In butterfly [1], in breaststroke [3], in backstroke [2], in freestyle [4] and in individual medley [11], women were able to reduce the gap to men in different age groups. In 3000-m open-water swimming, however, women were not able to reduce the sex difference to men [14]. Consequently, the existing sex difference regarding all swimming strokes is evident during the youngest (i.e., 25 to 29 years) age groups, while for nearly all the rest of the age groups (i.e., 30 years and older), women tend to reduce this sex gap.

In addition, for youth swimmers, age is of importance. When boys and girls from the age of 5 to 18 years for 50 m to 1500 m from the all-time top 100 *U*.S. freestyle swimming performances were investigated, the top five girls were faster until the age of ~10 years than boys. After the age of 10 years, however, boys were increasingly faster than girls until the age of ~17 years [43]. Overall, female swimmers can beat male swimmers under the age of ~10 years and achieve almost the same performance as men in the highest age groups (i.e., older than ~75–80 years) depending upon the distance and the stroke.

A few studies have investigated the age effect in open-water swimming. The age of peak performance increased over calendar years in long-distance open-water swimming. In the ’Manhattan Island Marathon Swim’, the age of the annual three fastest swimmers increased between 1983 and 2013 from ~28 to ~38 years for women and from ~23 to ~42 years for men [38]. In the 26.4 km open-water ultra-swim ’Marathon Swim in Lake Zurich’, Switzerland, the mean age of the finishers during the period 1987–2011 was ~32.0 years for men and ~30.9 years for women. The mean age of finishers and the age of winners increased across the years for both sexes [9].

### 3.5. Sex Difference and Swimming Strokes

Some studies investigated the aspect of sex difference for different strokes. In pool-swimming competitions, athletes perform in the four strokes (i.e., butterfly, backstroke, breaststroke and freestyle) [28] as well as in the combination of all four strokes as individual medley [11]. There seem to be changes in the sex difference for the swimming strokes depending upon the distance and the performance level. For both 200-m and 400-m freestyle and individual medley, no sex difference was found between neither the two distances, nor between the two swimming strokes [44]. The sex differences were ~9.7% and ~7.1% in individual medley and ~10.1% and ~6.1% in freestyle, respectively [45]. For elite male and female butterfly and freestyle swimmers at national level, the sex difference in peak swimming speed was lower in butterfly than in freestyle [31,46]. For national and international breaststroke and freestyle swimmers, the sex differences in swimming speed increased over time for national swimmers, but not for international swimmers for freestyle, while the sex difference remained stable for both national and international breaststroke swimmers [30]. The disparate findings were explained by the different performance levels, the different distances and strokes and the different sample sizes.

### 3.6. Performance Level and Sex Difference in Performance

In some studies, the changes in sex difference over time were investigated for different levels of athletes (i.e., national level, international level) [44,47]. In swimmers competing at national and international level, the sex-related difference in swimming speed was greater for freestyle than for breaststroke in 50-m to 200-m race distances for national swimmers, but not for international swimmers. For both groups, the sex-related difference for both freestyle and breaststroke swimming speeds decreased with increasing race distance. The sex-related differences in performance were greater for freestyle than for breaststroke for swimmers at national level, but not for swimmers at international level [47]. The disparate findings were explained by differences in performance level, distance, stroke and sample size.

### 3.7. Changes in Swimming Performance Over Years

Some studies investigated the change in performance over calendar years [8,48]. There seem to be differences between sexes, disciplines, performance level and distances [8]. In the past, it has already been suggested that women would soon perform better than men in swimming. In 1977, it was reported that women were gaining on their male counterparts at the rate of 0.45% a year in the 100-yard freestyle [48]. It was assumed that with that rate of improvement national level women may catch up with male counterparts by the year 2003. Likewise, in the 1650-yard freestyle, women were gaining on men, but at a slower rate of improvement of ~0.155%. It was assumed that it would take the women ~51 years to catch up to the men. The authors found that race times in women were improving at a rate faster than race times in men, but at some time in the future the rate of growth would probably stabilize for both sexes [48]. While at present that assumption, regarding the specific race distance (i.e., 100 yards), has not been completely fulfilled, it remains to be seen if a performance plateau would allow women to outperform men.

Regarding newer studies, female performance has improved over calendar years in some instances [8]. These analyses of changes in performance over the years have been performed for elite pool swimmers [5], for open-water swimmers [7,8] and for master swimmers competing in freestyle [4], backstroke [2], butterfly [1], breaststroke [3], individual medley [11] and in 3000-m open-water swimming [14]. In pool-swimming, performance was improved for most distances in both elite and master swimmers in backstroke [49], freestyle [31,46], breaststroke [30], butterfly [1,50], individual medley [11,30] and in 3000-m open-water swimming [14].

Some studies have investigated open-water swimming and showed that performance changed over years. For women and men crossing the ’Catalina Channel’ between 1927 and 2014, performance decreased nonlinearly in the annual fastest men and women [36]. In the ’Manhattan Island Marathon Swim’, race times of the annual three fastest women and men did not differ between sexes and remained stable across the years [38]. In the ’Maratona del Golfo Capri-Napoli’, race times of the annual fastest swimmers decreased linearly for women and for men from 1954 to 2013 from 39.2% to 4.7% [51]. For the annual top three swimmers, race times decreased linearly between 1963 and 2013 for women and for men from 38.2% ± 14.0% to 6.0% ± 1.0% [51]. In the ’English Channel Crossing’, the performance of the annual top three swimmers showed no changes either both females or males over the last 36 years and the sex difference remained unchanged at ~12.5% over the years [7]. In the ’English Channel Crossing’, performance increased progressively for both sexes, but was lower for female than for male athletes from 1900 to 2010 [8].

A different kind of events was the FINA races which were not held as solo events and swimmers could draft. For elite male and female swimmers competing at the FINA World Cup events of 5 km, 10 km and 25 km events, swimming speed of the annual ten fastest women decreased at 5 km and at 25 km, while it increased at 10 km. For the annual ten fastest men, peak swimming speed decreased at 5 km, while it remained unchanged at both 10 km and 25 km [10]. In the FINA 10 km competitions (i.e., World Cup races, European Championships, World Championships and Olympic Games) held between 2008 and 2012, swimming speed of the fastest women and men showed no changes across the years. Performance of the top ten female swimmers per event remained stable across calendar years. The top ten male swimmers per event showed a decrease in performance over years, even though swimming speed in the first race (i.e., January 2008, 1.40 m/s) was slower than swimming speed in the last race (i.e., October 2012, 1.50 m/s) [50]. The disparate findings were explained by the different performance levels, the different distances and strokes, the different ages, the different periods of time and the different sample sizes.

### 3.8. Swimming in Multi-Sports Disciplines Like Triathlons

Swimming is the first segment of a triathlon event, followed by cycling and then running [52]. Several studies investigated the trends in performance and the sex difference in performance in swimming in triathlons of different lengths such as the Olympic distance triathlon (i.e., 1.5  km swimming, 40  km cycling and 10  km running) [16,17,53], the Ironman distance triathlon (i.e., 3.8 km swimming, 180 km cycling and 42.195 km running) [16] and ultra-triathlon distances longer than the Ironman distance [20,54,55].

In Olympic distance triathletes competing in the ’Zürich Triathlon’ in Switzerland from 2000 to 2010, the sex difference in swimming was 15.2% for the top five triathletes overall [17]. For the world’s best triathletes at the ITU (International Triathlon Union) World Triathlon Series during the 2009–2012 period including the 2012 London Olympic Games, swim times and the sex difference in swimming remained unchanged [53].

For longer triathlon distances than the Olympic distance, sex difference has been investigated for the Ironman distance [19,56,57] and longer triathlon distances from 2× to 10× the Ironman distance [20,57]. It was shown that women improved swimming performance and closed the gap to men. In ’Ironman Hawaii’, the overall top ten men finishers improved their swimming performance between 1983 and 2012. The sex difference remained unchanged over the years at ~12.5% [19]. For the annual three best finishers in ’Ironman Hawaii’, the sex difference decreased nonlinearly in swimming between 1978 and 2013 [57]. In ’Isklar Norseman Xtreme Triathlon’ held over the Ironman distance, athletes swim at a water temperature of ~13–15 °C. Men were faster than the women in cycling, but not in swimming, running or overall race time. Across years, women improved their performance in swimming and both women and men improved their performance in cycling and in overall race time. In running, however, neither women nor men improved [3].

Different findings were, however, reported for longer triathlon distances. In Double Iron ultra-triathlon (i.e., 7.6 km swimming, 360 km cycling and 84.4 km running), men (2:36 h:min) were ~8 min faster than women (2:44 h:min) [20]. For triathlon distances from the Ironman distance in ’Ironman Hawaii’ to the Double Deca Iron ultra-triathlon distance (i.e., 76 km swimming, 3600 km cycling and 840 km running), the sex difference in performance showed no change with increasing race distance with the exception for the swimming split where the sex difference increased with increasing race distance for the three fastest ever [57].

Regarding triathlon swimming performance, in most cases, the sex difference tends to remain unchanged over the years. However, as previously mentioned, women can outperform men in specific triathlon races under more extreme conditions (i.e., water temperature of ~13–15 °C).

### 3.9. The Change in Sex Difference Over Years

In the same way where swimming performance can change over years, also the sex difference in swimming performance may change over years [10,44,58,59]. In some instances, women reduced the gap to men [59,60], in others not [10,28,44,53]. In these studies, pool-swimmers of sub-elite and elite level [28,44], open-water long-distance swimmers [59], master swimmers up to very high ages and swimmers in triathlons [20,54,55] were analyzed. In pool-swimming, the changes in sex difference differ over time regarding the distance, the age and the discipline [14,28,31,47]. According to this information, the sex difference in relation to pool-swimming and open-water swimming performance is largely dependent on the parameters analyzed in this review (i.e., competition level, swimming stroke and distance).

Several studies have examined the variation in sex difference over calendar years in open-water swimming events. In some events, the sex difference decreased, and the women reduced the gap to men [32,39,56], but in others not [9]. In women and men crossing the ’Catalina Channel’, the sex difference for all women and men decreased linearly between 1927 and 2014 from 52.4% to 7.1% [36]. The decrease of sex difference was linear suggesting that women continuously reduced the sex difference to men [36]. In the 36 km ’Maratona del Golfo Capri-Napoli’, the sex difference for the annual fastest swimmers, decreased linearly from 39.2% to 4.7% from 1955 to 2013 [51]. For the annual three fastest swimmers, the sex difference in performance decreased linearly from 38.2% ± 14.0% to 6.0% ± 1.0% from 1963 to 2013 [51]. Again, in this event, the linear change in both race times and sex differences indicates that women could achieve men’s performance or even to perform better than men in the near future in this event [51]. In ’La Traversée Internationale du Lac St-Jean’ (32 km) held between 1955 and 2012 in Canada, the sex difference remained unchanged over years for the annual fastest women and men at 8.8% [58]. For the annual three fastest women and men, the sex difference decreased across years (1975–2011) from 14.4% ± 11.0% to 3.7% ± 1.4% [58]. Overall, most studies found that women reduced the gap to men over years in open-water swimming.

Differences were found in open-water swimming where women are allowed to draft behind men. For elite male and female swimmers competing in 5 km, 10 km and 25 km open-water FINA World Cup races, elite female swimmers improved their performance in 10 km, but impaired performance in 25 km, leading to a linear decrease in sex difference in 10 km and a linear increase in sex difference in 25 km. The linear change in sex differences suggests that women will improve in the near future in 10 km, but not in 25 km [10]. In elite open-water swimmers competing at FINA 10 km races, the mean sex difference in performance for the fastest swimmers was stable across years [50].

In long triathlon races, women were not able to close the gap to men. For triathlon races longer than the Ironman race distance, the sex difference in swimming showed no change over years in either Double Iron ultra-triathlon [54] or in ’Ultraman Hawaii’ [55]. In Double Iron ultra-triathlon races, the swimming times remained unchanged across years with an unchanged sex difference for the annual three fastest women and men [20]. Potential explanations as to why sex differences decreased over the years or not could be the selected period of time [44,53,58], the level of the investigated athletes (i.e., annual fastest, annual ten fastest, national level, international level, etc.) [44,58], and/or whether the swimmers were solo swimmer, competing in a drafting race or triathletes. When the sex difference showed no change over time, the investigated period of time was most likely too short [53].

### 3.10. The Influence of Swimwear on Performance

Sex differences among swimmers using a wetsuit have also been investigated [61]. A wetsuit can be mainly used in the swim split in triathlon races [61] or in long-distance open-water swimming races [62]. In triathletes, the effect of a wetsuit on lighter female swimmers was no different than the effect on heavier male swimmers [61]. Swimming with or without a wetsuit shows a difference between the sexes [63,64]. When swimming speed was compared among women and men with or without wetsuit over different distances, wearing a wetsuit improved swimming speed for both women and men, but the benefit of the use of wetsuits depends on additional factors such as race distance. Women may be favored from wearing a wetsuit more than men in longer ultra-distance races of open-water swimming [63]. It has also been shown that high-tech swimsuits gave more pronounced advantage to men than women and for low resistance as compared with high resistance swimming strokes [64]. Yet, the new rules on swimsuits, in effect since 2010, should also be considered when analyzing swimming performance.

## 4. Potential Explanations for The Female Dominance in Long-Distance Open Water Swimming and Age Group Swimmers

Based on these results, we may conclude that women were able to outperform men in swimming in solo, long-distance swimming events held in water temperatures between 15 °C and 20 °C. Women were also able to achieve the same performance as men in all distances and disciplines of pool swimming at younger ages (i.e., younger than ~10 years) and older ages (i.e., from ~80 years onwards).

Two main variables may explain why women can outperform men in open-water swimming: (i) the long distance of ~30 km, (ii) and water colder than ~20 °C. Potential explanations for the finding that women can achieve a better performance than men can be attributed to differences in anthropometric characteristics such as body composition [65], body weight [66], body fat [67,68,69,70], lean body mass [71,72], body height [73,74], muscle thickness [75] and muscle size [72]. Other possible explanations were differences in swimming biomechanics such as kinematic parameters [76], arm coordination and arm–leg coordination [76,77,78], energetic cost, differences in swimming economy and swimming efficiency [79,80,81], gliding [82], body roll [83], shoulder flexibility [84], trunk flexibility [85], knee flexibility [86,87], propelling efficiency [88]. Beyond that, additional differences in motivational aspects [89,90,91,92,93,94,95,96], physiology [97,98,99,100] and biochemistry [101,102,103], recovery [104] and injury prevalence [105,106] can be pointed out. We found, however, no potential explanation for the youth swimmers.

A very likely explanation that women were faster than men in specific open-water ultra-distance events is the fact that female swimmers have more body fat than male swimmers [67] leading to better insulation against the cold and better buoyancy for long swimming distances [69]. It is well known that sex is associated with waist-to-hip ratio and body fat percentage in swimmers [107]. Due to the higher body fat, female swimmers have a different body shape compared to male swimmers. Male swimmers have a more central distribution of fat when compared to females, where body fat is built up in the region of legs [108]. In nonstationary swimming with changing velocity, water around the swimmer is set in motion which can be thought of as an added mass of water. Female swimmers have a lower added mass and relative added mass than male swimmers suggesting that sex differences in body shape may be associated with added mass [109].

In ultra-distances of open-water swimming, different anthropometric characteristics such as body height, body mass index (BMI), length of arm and training characteristics (e.g., swimming speed) were associated with performance for men. For women, swimming speed during training was associated with performance, but not anthropometric characteristics. Considering all variables for men, BMI and swimming speed during training were related to race time, but not for women [110]. Differences in anthropometric characteristics do exist between female and male swimmers [111,112]. It was shown that differences in height, arm span, skinfold thicknesses (e.g., triceps, subscapular, crista iliaca, ileo-spinal, abdominal, thigh, leg, sum of skinfolds), bi-acromial-bi-iliac index, bone body mass, muscle and fat, ectomorphy and endomorphy exist [111]. In another study, elite female youth swimmers had greater skinfolds at triceps, suprailiac and abdominal site. Endomorphic somatotype was twofold greater among elite female compared to elite male youth swimmers [112].

Although women could outperform men in certain swimming disciplines, in general, elite men were faster than elite women [113]. The sex gap in swimming performance seemed to remain stable in shorter distances. In Olympic Trial swimming from 1972 to 2016, the performance gap in swimming remained at ~8% [114]. The plateau during these ~40 years in the performance gap highlighted the role of biologic background (e.g., longer limbs, larger muscle mass, greater aerobic capacity and lower fat mass) on race time. Current evidence indicates that women will not swim as fast as men in Olympic events, which justifies sex segregation in these individual sport disciplines [114]. Men have an advantage of larger body size and muscle mass, a superior ventilation function and anaerobic and aerobic energy transfer systems. It is well known that male swimmers have a higher maximum oxygen uptake than female swimmers among both younger [97] and older age groups [98,99,100]. It is also known that ventilation functions including forced vital capacity (FVC), forced expiratory volume in one second (FEV1), FEV1/FVC and mandatory minute ventilation (MMV) were superior in male athletes to those in females [115]. Furthermore, the average diameter of muscle fibers was larger in men than in women [116]. Therefore, it is not a surprise that male youth swimmers show increased power values in both their legs and arms [117]. Finally, although male swimmers have a higher muscle mass, maximum handgrip isometric strength values correlated with swimming race time, especially in female swimmers [118]. Compared to men, women have an enhanced ability to oxidize fat, superior hydrodynamics and more even pacing, which provide advantage, especially during prolonged swimming [113]. Regarding pacing in pool- swimming, an effect of sex on lap time in master swimmers competing in 100, 200, 400 and 800-m freestyle at the World Championships suggested greater changes of pacing in women than in men [12].

Apart from anthropometric and/or physiological differences, psychological differences may also explain female dominance in certain swimming disciplines [91,92,93,94,95,119,120,121]. Female swimmers show differences to male swimmers regarding mental toughness [92]. Competitive female swimmers were emotionally secure, physically healthy and reasonably contented with their present social status. For these women, the emotional, social, and physical costs were definitely worth the sacrifices [120].

Effects of cultural and sociopolitical norms and outdated stereotypes (i.e., reduced opportunity to participate and compete in sports) that influence the number of female competitors, particularly in earlier years, should also be considered. Women have traditionally been under-represented in sports. In the 19th century, women were engaged in non-competitive recreational activities, but not in competitive sport. In 1971 in the United States, less than 7% of high school varsity athletes were female. Title IX was set in the United States in 1972 with the aim of providing equal treatment in sports, regardless of sex and increasing the number of women in sports [122]. Even as recently as 1979 in Brazil, it was illegal for women to play football [123]. According to Capranica et al., there were still some countries in the 2012 London Olympic Games that did not have a female in their delegation [124]. There was an increase in the number of female athletes in all sports over the last century, including swimming, that likely contributed to the reduction in the gap in performance between the sexes [125].

A potential explanation for the improved performance of older women, especially in pool-swimming, is the fact that the age of peak performance has increased in women since the 1980s. When 116 years from the first Olympic Games (1898) to the 2014 Olympic Games were analyzed, regarding the ages at which peak performance was observed, peak performance ages in women have increased consistently since the 1980s in all the athletic events examined (i.e., track and field, swimming, rowing and ice-skating events). When the age of peak female performance increased, it became similar to the age of male performance in many events. In the last 20 to 30 years, the age of peak female athletic performance increased, but not the age of male athletic performance [126]. Age in open-water swimmers may also be of importance. In the ’Manhattan Marathon Island Swim’, the age of peak performance of ultra-distance swimmers has changed across the last decades, with the fastest swimmers getting older between 1983 and 2013. During this period, the age of the three fastest swimmers raised from 28 to 38 years in women and from 23 to 42 years in men [38]. The fact that female competitors were younger than their male counterparts likely had an effect on performance. Age was also of importance in other endurance athletes such as ultra-marathon runners [127,128]. In recent studies investigating ultra-marathoners [128] and athletes competing in tower running [129], women were able to close the gap to men [128] or to even outperform men [130]. In 50- and 100-mile ultra-marathoners, the sex difference in running performance decreased with increasing age and was smaller in the longer (100 miles) compared to the shorter (50 miles) distance [128]. This finding may be explained by the lower participation of women in longer ultra-marathon races. Also, in tower running (stair climbing), women were able to beat men in specific situations, e.g., in smaller buildings with less than 600 stairs for younger (30–59 years) and older (>69 years) age groups, in buildings with 1600–220 stairs for older ages (>69 years) and in buildings with more than 2200 stairs for younger (<20 years) and older (60–69 years) age groups [130]. With increasing age, experience may improve and race tactics may become better.

A further aspect was the use of technical wetsuits in pool-swimming. In the 2009 FINA World Championship held in Rome, a total of 43 world records were set. Men set new world records in 15 of those events, whereas women did the same in 17 events. Each of the men’s world records and 14 of the 17 women’s records still stood. In the past, these world records had not been broken in such a short period of time. There was much speculation that full-body, polyurethane, technical swimsuits were the reason for the improvement in world records. Further analysis led the FINA to institute new rules on 1st January 2010, that limited the types of technical swimsuits that could be worn by athletes. No long-course world record has been broken since then [129]. A problem in this field was, however, the fact that most of the considered studies used different numbers of subjects, different populations, different analyses leading to different significance levels. This problem cannot be eliminated, but it must be considered that depending on the sample and the analysis used, a significant difference between female and male performance can result or not.

## 5. Conclusions

The collective data presented in this review indicate that existing sex differences in swimming performance showed a generally diminishing trend that was more profound during the longer pool-races, for all swimming strokes. Age-related variations were also reported in both pool and open-water swimming, as sex difference mainly remained for the younger age groups. However, female athletes in very young age groups (10 years and younger) and very old age groups (75–80 years and older) outscored their male counterparts. When analyzing triathlon swimming performance, the sex gap remained stable during shorter and longer swim distances, except for events under more extreme water temperatures where women can outperform men. Finally, the sex difference in pool-swimming performance over time showed variations depending on the swimming stroke, distance and the competitive level. Regarding open-water swimming over time, women seemed to continuously narrow the gap to men, especially in specific long-distances, where the assumption of outperforming men existed. In summary, women tended to decrease the existing sex differences in specific age groups (i.e., younger than 10 years and older than 75–80 years) and swimming strokes or even overperform men in long-distance open-water swimming (distance of around 30 km), especially under extreme weather conditions (water colder than ~20 °C). Future studies may investigate body compositions of the different age group swimmers in order to explain the sex differences for specific sports disciplines such as pool-swimming and long-distance open-water swimming.

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
