# Peer review of "Sex Differences in Swimming Disciplines—Can Women Outperform Men in Swimming?"

_ijerph, 2020, doi:10.3390/ijerph17103651_

Round 1

Reviewer 1 Report

The manuscript is quite interesting, consolidating the body of knowledge on sex gap in swimming performance. However, upon reading this draft I would like to raise a few questions and remarks that hopefully will help to improve the authors´ work:

  1. Please provide more details on this work under abstract.
  2. Was there any systematic methodology selected to search and retrieve the manuscripts included in this narrative review? If not, what was the criteria to pick up the studies?
  3. Kindly elaborate on the rational for this study. I.e., why is there a growing interest on understanding the gender gap? E.g., is this also a trendy topic in other sports, beyond swimming such as other time-based sports?
  4. Authors are encouraged, as much as possible, to set the links between studies. I.e. at least based on my understanding, big chunks of text are just the listing of findings of experimental studies. But scarce effort has been put providing a critical analysis of such findings
  5. How much did the swim gear used in 2008-2009 has affected the sex gap in that period of time, but also has spilled until today? For instance, we do know that in 2019-2020 swimming community was still struggling to break work records set in 2008-2009. How did this phenomenon have impact on swimming performance and sex gap since then? Some deep elaboration on this would be appreciated.

Author Response

Reviewer 1

The manuscript is quite interesting, consolidating the body of knowledge on sex gap in swimming performance. However, upon reading this draft I would like to raise a few questions and remarks that hopefully will help to improve the authors´ work:

Please provide more details on this work under abstract.

Answer: We changed the abstract to ‘In recent years, the interest of female dominance in long-distance swimming grew where several newspapers articles were published speculating about the female performance and dominance especially in open-water ultra-distance swimming. The aim of this narrative review was to examine the difference between the sexes for all swimming strokes (i.e., butterfly, backstroke, breaststroke, freestyle and individual medley), different distances (i.e., from sprint to ultra-distances), extreme conditions (i.e., cold water), different ages, and swimming integrated in multi-sports disciplines, such as triathlon, in various age groups and over calendar years. The influence of various physiological, psychological, anthropometrical and biomechanical aspects is also discussed. The data bases Scopus and PUBMED were searched by April 2020 for the terms and ‘sex – difference - swimming’. In open-water long-distance swimming events of the 'Triple Crown of Open Water Swimming' with the 'Catalina Channel Swim', the 'English Channel Swim' and the 'Manhattan Island Marathon Swim', women were about 0.06 km/h faster than men. Two main variables may explain why women can swim faster than men in open-water swimming events: (i) the long distance of around30 km, (ii) and water colder than ~20 °C. Recent studies investigated the performance trends and the sex difference in performance in master swimmers (i.e., age groups 25-29 to 90-94 years) competing between 1986 and 2014 in the FINA (Fédération Internationale de Natation) World Championships in pool swimming in freestyle, backstroke, butterfly, breaststroke, individual medley and in 3000 m open-water swimming. Women seemed able to achieve almost the same performance as men in the oldest age groups (i.e., older than 75-80 years). In boys and girls aged 5-18 years and listed in the all-time top 100 U.S. freestyle swimming performances from 50 m to 1500 m, the five fastest girls were faster until the age of about 10 years than the five fastest boys. After the age of 10 years until the age of 17 years, however, boys were increasingly faster than girls. Therefore, women tend to decrease the existing sex differences in specific age groups (i.e., younger than 10 years and older than 75-80 years) and swimming strokes or even overperform men in long-distance open-water swimming (distance of around 30 km), especially under extreme weather conditions (water colder than ~20 °C). Future studies might investigate more detailed the very young (<10 years) and very old (>75-80 years) age groups in swimming’.

Was there any systematic methodology selected to search and retrieve the manuscripts included in this narrative review? If not, what was the criteria to pick up the studies?

Answer: We added a method section with ‘In recent years, the interest of female dominance in long-distance swimming grew where several newspapers articles were published speculating about the female performance and dominance especially in open-water ultra-distance swimming. In one article, the history of female performance in open-water swimming starting with Gertrud Ederle in the English Channel Crossing was actualized and the female dominance in Manhattan Island Marathon Swim. The author discussed the problem of comparing the fastest men and women in contrast to all men and women, leading to a different finding regarding male or female dominance [https://explorersweb.com/2019/07/05/why-women-excel-at-marathon-swimming/]. Also, another article describing the sex difference in swimming and running discussed the aspect of the fastest women and men [https://www.thecut.com/2016/09/the-obscure-endurance-sport-women-are-quietly-dominating.html]. Steven Munatones, one of the world’s top experts on open-water swimming, found that the average female time was 33 min faster than the average male time in the 135 years of the English Channel Crossing [https://womenintheworld.com/2016/09/18/the-grueling-sport-in-which-women-competitors-typically-outperform-the-men/]. It seems that women are better in long-distance open-water swimming. For example, the female swimmer Sarah Thomas was the first person in the world to swim the English Channel four times in a row without stopping [https://www.theguardian.com/lifeandstyle/2020/jan/03/female-ultra-athletes-leading-field-women-less-ego]. In a further article, women were described winning ultra-endurance races in cycling and running ahead of all men [https://www.bbc.com/news/world-49284389]. Furthermore, the fastest women ever were faster than the fastest men ever in the Catalina Channel Swim and in the Manhattan Island Marathon Swim [https://www.healthline.com/health-news/will-women-athletes-ever-be-able-to-compete-with-men#2]. The data bases Scopus and PUBMED were searched by April 2020 for the terms and ‘sex – difference - swimming’. The search in Scopus lead to 239 entries, the search in PUBMED to 558 entries. We excluded case studies, case reports, animal studies, studies with divers and rowers, and studies with patients. Although the newspapers articles primarily reported about open-water long-distance swimming, we consider in this narrative review all scientific results upon differences between the sexes in swimming for all swimming strokes (i.e., butterfly, backstroke, breaststroke, freestyle and individual medley), distances (i.e., from sprint to ultra-distances), conditions (i.e., cold water), ages, and swimming integrated in multi-sports disciplines such as triathlon

Kindly elaborate on the rational for this study. I.e., why is there a growing interest on understanding the gender gap? E.g., is this also a trendy topic in other sports, beyond swimming such as other time-based sports?

Answer: We added a method section with ‘In recent years, the interest of female dominance in long-distance swimming grew where several newspapers articles were published speculating about the female performance and dominance especially in open-water ultra-distance swimming. In one article, the history of female performance in open-water swimming starting with Gertrud Ederle in the English Channel Crossing was actualized and the female dominance in Manhattan Island Marathon Swim. The author discussed the problem of comparing the fastest men and women in contrast to all men and women, leading to a different finding regarding male or female dominance [https://explorersweb.com/2019/07/05/why-women-excel-at-marathon-swimming/]. Also, another article describing the sex difference in swimming and running discussed the aspect of the fastest women and men [https://www.thecut.com/2016/09/the-obscure-endurance-sport-women-are-quietly-dominating.html]. Steven Munatones, one of the world’s top experts on open-water swimming, found that the average female time was 33 min faster than the average male time in the 135 years of the English Channel Crossing [https://womenintheworld.com/2016/09/18/the-grueling-sport-in-which-women-competitors-typically-outperform-the-men/]. It seems that women are better in long-distance open-water swimming. For example, the female swimmer Sarah Thomas was the first person in the world to swim the English Channel four times in a row without stopping [https://www.theguardian.com/lifeandstyle/2020/jan/03/female-ultra-athletes-leading-field-women-less-ego]. In a further article, women were described winning ultra-endurance races in cycling and running ahead of all men [https://www.bbc.com/news/world-49284389]. Furthermore, the fastest women ever were faster than the fastest men ever in the Catalina Channel Swim and in the Manhattan Island Marathon Swim [https://www.healthline.com/health-news/will-women-athletes-ever-be-able-to-compete-with-men#2]. The data bases Scopus and PUBMED were searched by April 2020 for the terms and ‘sex – difference - swimming’. The search in Scopus lead to 239 entries, the search in PUBMED to 558 entries. We excluded case studies, case reports, animal studies, studies with divers and rowers, and studies with patients. Although the newspapers articles primarily reported about open-water long-distance swimming, we consider in this narrative review all scientific results upon differences between the sexes in swimming for all swimming strokes (i.e., butterfly, backstroke, breaststroke, freestyle and individual medley), distances (i.e., from sprint to ultra-distances), conditions (i.e., cold water), ages, and swimming integrated in multi-sports disciplines such as triathlon

Authors are encouraged, as much as possible, to set the links between studies. I.e. at least based on my understanding, big chunks of text are just the listing of findings of experimental studies. But scarce effort has been put providing a critical analysis of such findings

Answer: We try to improve this aspect. We structure the manuscript now with the main findings and then with the potential explanations. We also add a section A problem in this field is the fact that most of the considered studies used different numbers of subjects, different populations, different analyses leading to different significance levels. This problem cannot be eliminated, but is must be considered that depending of the sample and the analysis used, a significant difference between female and male performance can result or not’ at the end of the discussion before the conclusions.

How much did the swim gear used in 2008-2009 has affected the sex gap in that period of time, but also has spilled until today? For instance, we do know that in 2019-2020 swimming community was still struggling to break work records set in 2008-2009. How did this phenomenon have impact on swimming performance and sex gap since then? Some deep elaboration on this would be appreciated.

Answer: We consider this aspect in a separate section with ‘A further aspect is the use of technical wetsuits. In the 2009 FINA World Championship held in Rome, a total of 43 world records were set. Men set new world records in 15 of those events, whereas women did the same in 17 events. Each of the men's world records and 14 of the 17 women's records still stand. Never before had these many world records been broken in such a short period of time. There was much speculation that full-body, polyurethane, technical swimsuits were the reason for the improvement in world records. Further analysis led the FINA to institute new rules on January 1, 2010, that limited the types of technical swimsuits that could be worn by athletes. No long-course world record has been broken since then’.

Reviewer 2 Report

I am impressed by the amount of literature review done by the authors, but unfortunately, I must say that is the only strength of this manuscript. The authors cited so many articles, but what they did was just listing every result in the articles without summarising it. There are so many contradicting results out there in the ocean of academic studies, and I think the role of a review paper is to pick up useful information pieces and make a strong link between them. The authors did find relevant articles, but they did not make links between them and summarise key points, and that made the manuscript extremely difficult to follow since there is no flow in the text. For example, in L432, the authors stated that "Based on these results, we may conclude that women are able to outperform men in swimming in solo...", I strongly doubt if the authors had to use over 10 pages with 60 references to reach this conclusion. Judging from what authors wrote up to this point (L432), I think around 50 articles are not necessary. 

Furthermore, since the authors just listed everything without examining the findings in each article carefully, there are so many conflicting statements. One very good example is L 502-512. I probably do not even have to explain what is wrong here. If the authors (or anybody who does not even have an academic background) read this paragraph once again, they should be able to realise that this paragraph makes no sense. 

Again, I think the amount of reading work the authors did is impressive, but a complete re-writing would be necessary to make their work in a publication in my opinion. 

Author Response

Reviewer 2

I am impressed by the amount of literature review done by the authors, but unfortunately, I must say that is the only strength of this manuscript. The authors cited so many articles, but what they did was just listing every result in the articles without summarizing it. There are so many contradicting results out there in the ocean of academic studies, and I think the role of a review paper is to pick up useful information pieces and make a strong link between them.

Answer: We try to improve this aspect. We structure the manuscript now with the main findings and then with the potential explanations.

The authors did find relevant articles, but they did not make links between them and summarize key points, and that made the manuscript extremely difficult to follow since there is no flow in the text. For example, in L432, the authors stated that "Based on these results, we may conclude that women are able to outperform men in swimming in solo...", I strongly doubt if the authors had to use over 10 pages with 60 references to reach this conclusion. Judging from what authors wrote up to this point (L432), I think around 50 articles are not necessary. 

Answer: In Line 432 we start with potential explanations of the presented findings.

Furthermore, since the authors just listed everything without examining the findings in each article carefully, there are so many conflicting statements. One very good example is L 502-512. I probably do not even have to explain what is wrong here. If the authors (or anybody who does not even have an academic background) read this paragraph once again, they should be able to realize that this paragraph makes no sense. 

Answer: We changed that section to ‘Apart from anthropometric and/or physiological differences, psychological differences might also explain female dominance in certain swimming disciplines [93-97,121,122]. Female swimmers show differences to male swimmers regarding mental toughness [94]. Competitive female swimmers are emotionally secure, physically healthy, and reasonably contented with their present social status. For these women, the emotional, social, and physical costs were definitely worth the sacrifices [122].

Again, I think the amount of reading work the authors did is impressive, but a complete re-writing would be necessary to make their work in a publication in my opinion. 

Answer: We re-wrote a large part of the manuscript and included all details of the tables in the text in order to eliminate a potential bias in self-citation.

Reviewer 3 Report

This narrative review provides an in-depth overview of various scientific works on sex differences in swimming. The authors combined many different perspectives on where sex differences occur, how they change over time & with age, and how they might be explained. A fascinating work, but I have to raise some serious concerns with the scientific value of the manuscript.

Unfortunately, this narrative review gives a strong sense of bias. The most obvious reason for this feeling of bias is the large portion of the reviewed work that was done by the authors of this review. This is in itself not a reason for bias, given that indeed this research group has produced by far the most work on this topic, but it does mean that the authors should be extra careful to avoid bias. In my opinion, the authors have not at all managed to avoid such a bias.

This ‘air of bias’ is a result of:

1) the prolific self-referencing

Referencing your own work as often as you do can be problematic. Especially when zooming in on the tables: all references in the tables are to your own group’s work. It is understandable that your own work occurs so often in the literature, as your group has done so much work in this topic, however, it makes it so much more difficult to come up with an unbiased narrative.

2) the lack of critical assessment of the referenced works (including their own work)

The first point of critique is aggravated by this second point. In your narrative review, you never seem to critique, doubt, question, any of the findings you discuss. This is problematic, as there are always considerations that have to be taken into account when comparing specific findings (e.g., the number of participants, the type of population, the analysis method, the significance level, etc.).

Let me be more specific:

Often, statements are simply taken from the papers without any sense of doubt:

“Age is also of importance in other endurance athletes such as ultra-marathon runners [135,136].”

--> How many people were included? What were the age ranges? Who weren’t included? How long was the history of the data?

“Potential explanations for the finding that women can achieve a better performance than men can be attributed to differences in anthropometric characteristics such as body composition [61], body weight [62], body fat [63-66], lean body mass [67,68], body height [69,70], muscle thickness [71], and muscle size [68]."

--> What do these differences explain? How were these claims made? How big are the differences for the general population? Is there a reason to believe that these differences are exaggerated for the (long distance) swimmers?

Line 142. The authors claim that “the difference reached no statistical significance”.

--> This raises all sorts of questions: how can such a significance be assessed? Is it reliable? What does ‘significant’ mean in the context of looking 10 outliers (i.e., the 10 fastest women)?

3) lack of clarity search strategy and inclusion criteria

The only mention of the search strategy is at the end of the Introduction section: “The data bases Scopus and PUBMED were searched by March 2020 for the terms ‘swimming’ and ‘sex difference’.” How many results were found? How many were included? Why were some papers excluded? Why were only these search terms used? Did you look for ‘gender difference’? Why did you only look for ‘difference’? Other terms might also occur (e.g., bias, advantage).

And even if this was the only search strategy, why was this paper not included: Examining the relationship between sex and motivation in triathletes.

4) lack of clarity in the methodology (where do the tables come from, and why are you only including your own works in these summary tables)

And then the tables. They provide an interesting overview of many different studies, albeit all studies of the same research group. However, nowhere in the paper it is mentioned how these tables were made. Which data was included? How were the comparisons made? Was the data publicly available?

And finally, although it is impressive that the authors take such a broad approach, their storyline is all over the place:

  1. Introduction
  2. Sex difference regarding swimming distance in pool-swimming
  3. Open-water swimming
  4. Ice swimming
  5. Age
    1. Pool-swimming
    2. Open-water swimming
  6. Sex difference and swimming strokes
  7. Performance level and sex difference in performance
  8. Changes in swimming performance over years
  9. Swimming in multi-sports disciplines like triathlons
  10. The change in sex difference over years
    1. Pool-swimming
    2. Open-water long-distance swimming
    3. Swim split of longer triathlon races
  11. The influence of swimwear on performance
  12. Potential explanations for the female dominance
    1. Female dominance in long-distance swimming
    2. The influence of body fat on female swimming performance
    3. Other anthropometric and physiological characteristics of female swimming performance
    4. Psychological aspects and female swimming performance
    5. Kinematics and swimming performance
    6. Other aspects to be considered
  13. Conclusions

There are headers that concern types of swimming (pool, open water, ice, multi-sports disciplines), there are headers that discuss factors that influence performance (age, the rise of female sports over the years, performance level, swimwear) and finally, headers that discuss the explanations of the sex differences.

Specific Comments

Line 135: It seems that the unit of measurement is incorrect. Either way, please be consistent throughout the manuscript.

Line 426: space between “swimming” and “[59]”

In conclusion…

Overall, this manuscript feels more like a book chapter or a piece in a popular scientific journal. Very interesting to read, a nice synthesis of a large body of information, but, it does not contribute much to the domain of science. Unless the authors manage to provide a bias-free coherent manuscript, I would not deem it fit for publication.

Author Response

Reviewer 3

This narrative review provides an in-depth overview of various scientific works on sex differences in swimming. The authors combined many different perspectives on where sex differences occur, how they change over time & with age, and how they might be explained. A fascinating work, but I have to raise some serious concerns with the scientific value of the manuscript.

Answer: We tried to address all comments.

Unfortunately, this narrative review gives a strong sense of bias. The most obvious reason for this feeling of bias is the large portion of the reviewed work that was done by the authors of this review. This is in itself not a reason for bias, given that indeed this research group has produced by far the most work on this topic, but it does mean that the authors should be extra careful to avoid bias. In my opinion, the authors have not at all managed to avoid such a bias.

Answer: The data bases Scopus and PUBMED were searched by April 2020 for the terms and ‘sex – difference - swimming’. The search in Scopus lead to 239 entries, the search in PUBMED to 558 entries. When we take PUBMED and the key words, we get as mentioned 558 studies, see https://www.ncbi.nlm.nih.gov/pubmed/?term=sex+difference+swimming. When we use these key words and add the first author who published primarily in this topic, see https://www.ncbi.nlm.nih.gov/pubmed/?term=sex+difference+swimming+knechtle+b , we come to 49 studies. This is about 10% of the published work for the terms ‘sex – difference – swimming’.

This ‘air of bias’ is a result of:

1) the prolific self-referencing

Referencing your own work as often as you do can be problematic. Especially when zooming in on the tables: all references in the tables are to your own group’s work. It is understandable that your own work occurs so often in the literature, as your group has done so much work in this topic, however, it makes it so much more difficult to come up with an unbiased narrative.

Answer: We deleted all tables and included the results of the tables in the text.

2) the lack of critical assessment of the referenced works (including their own work)

The first point of critique is aggravated by this second point. In your narrative review, you never seem to critique, doubt, question, any of the findings you discuss. This is problematic, as there are always considerations that have to be taken into account when comparing specific findings (e.g., the number of participants, the type of population, the analysis method, the significance level, etc.).

Answer: We fully agree with the expert reviewer and add a section where we address this aspect. This is a problem in science since every author uses a different sample of subjects and different methods. We add this section ‘A problem in this field is the fact that most of the considered studies used different numbers of subjects, different populations, different analyses leading to different significance levels. This problem cannot be eliminated, but is must be considered that depending of the sample and the analysis used, a significant difference between female and male performance can result or not’ at the end of the discussion before the conclusions.

Let me be more specific:

Often, statements are simply taken from the papers without any sense of doubt:

“Age is also of importance in other endurance athletes such as ultra-marathon runners [135,136].”

--> How many people were included? What were the age ranges? Who weren’t included? How long was the history of the data?

Answer: This is an introducing sentence for a section to show that also in other sports disciplines such as ultra-marathon running women were able to close the gap to men as it was found in pool-swimming for master swimmers. We changed that specific section to ‘Age is also of importance in other endurance athletes such as ultra-marathon runners [141,142]. In recent studies investigating ultra-marathoners [142] and athletes competing in tower running [143], women were able to close the gap to men [142] or to even outperform men [143]. In 50- and 100-mile ultra-marathoners, the sex difference in running performance decreased with increasing age and was smaller in the longer (100 miles) compared to the shorter (50 miles) distance [142]. This finding might be explained by the lower participation of women in longer ultra-marathon races. Also, in tower running (stair climbing), women were able to beat men in specific situations, e.g. in smaller buildings with less than 600 stairs for younger (30-59 years) and older (>69 years) age groups, in buildings with 1600-220 stairs for older ages (>69 years) and in buildings with more than 2200 stairs for younger (<20 years) and older (60-69 years) age groups [143] ’ to make it easier to understand for non-experts in this topic.

“Potential explanations for the finding that women can achieve a better performance than men can be attributed to differences in anthropometric characteristics such as body composition [61], body weight [62], body fat [63-66], lean body mass [67,68], body height [69,70], muscle thickness [71], and muscle size [68]."

--> What do these differences explain? How were these claims made? How big are the differences for the general population? Is there a reason to believe that these differences are exaggerated for the (long distance) swimmers?

Answer: After the section ‘findings’ where we present the results of the literature, we try to explain the female dominance. We added some newspapers articles where journalists and experts made their considerations. Based on that, we looked for scientific articles and tried to find potential explanations.

Line 142. The authors claim that “the difference reached no statistical significance”.

--> This raises all sorts of questions: how can such a significance be assessed? Is it reliable? What does ‘significant’ mean in the context of looking 10 outliers (i.e., the 10 fastest women)?

Answer: In this field and in the scientific literature for this field, different scientists used different approaches. Some compared the trend in world records, some compared the best athletes (top 3, top 5, top 10, etc). Independent of the data set and the sample, one can make a comparison between women and men. However, depending upon the sample used, the result will differ. Based on that we included in the text which samples were compared to make it easier to understand.

3) lack of clarity search strategy and inclusion criteria

The only mention of the search strategy is at the end of the Introduction section: “The data bases Scopus and PUBMED were searched by March 2020 for the terms ‘swimming’ and ‘sex difference’.” How many results were found? How many were included? Why were some papers excluded? Why were only these search terms used? Did you look for ‘gender difference’? Why did you only look for ‘difference’? Other terms might also occur (e.g., bias, advantage).

Answer: We add a ‘method section’ with ‘In recent years, the interest of female dominance in long-distance swimming grew where several newspapers articles were published speculating about the female performance and dominance especially in open-water ultra-distance swimming. In one article, the history of female performance in open-water swimming starting with Gertrud Ederle in the English Channel Crossing was actualized and the female dominance in Manhattan Island Marathon Swim. The author discussed the problem of comparing the fastest men and women in contrast to all men and women, leading to a different finding regarding male or female dominance [22]. Also, another article describing the sex difference in swimming and running discussed the aspect of the fastest women and men [23]. Steven Munatones, one of the world’s top experts on open-water swimming, found that the average female time was 33 min faster than the average male time in the 135 years of the English Channel Crossing [24]. It seems that women are better in long-distance open-water swimming. For example, the female swimmer Sarah Thomas was the first person in the world to swim the English Channel four times in a row without stopping [25]. In a further article, women were described winning ultra-endurance races in cycling and running ahead of all men [26]. Furthermore, the fastest women ever were faster than the fastest men ever in the Catalina Channel Swim and in the Manhattan Island Marathon Swim [27].The data bases Scopus and PUBMED were searched by April 2020 for the terms and ‘sex – difference - swimming’. The search in Scopus lead to 239 entries, the search in PUBMED to 558 entries. We excluded case studies, case reports, animal studies, studies with divers and rowers, and studies with patients. Although the newspapers articles primarily reported about open-water long-distance swimming, we consider in this narrative review all scientific results upon differences between the sexes in swimming for all swimming strokes (i.e., butterfly, backstroke, breaststroke, freestyle and individual medley), distances (i.e., from sprint to ultra-distances), conditions (i.e., cold water), ages, and swimming integrated in multi-sports disciplines such as triathlon.

’ to explain the strategy.

And even if this was the only search strategy, why was this paper not included: Examining the relationship between sex and motivation in triathletes.

Answer: We checked again our records, but this mentioned study was, unfortunately, not among the records when searching the data bases as explained in the ‘method section’.

4) lack of clarity in the methodology (where do the tables come from, and why are you only including your own works in these summary tables)

Answer: We deleted all tables and included the results of the tables in the text.

And then the tables. They provide an interesting overview of many different studies, albeit all studies of the same research group. However, nowhere in the paper it is mentioned how these tables were made. Which data was included? How were the comparisons made? Was the data publicly available?

Answer: We deleted all tables and included the results of the tables in the text.

And finally, although it is impressive that the authors take such a broad approach, their storyline is all over the place:

  1. Introduction
  2. Sex difference regarding swimming distance in pool-swimming
  3. Open-water swimming
  4. Ice swimming
  5. Age
    1. Pool-swimming
    2. Open-water swimming
  6. Sex difference and swimming strokes
  7. Performance level and sex difference in performance
  8. Changes in swimming performance over years
  9. Swimming in multi-sports disciplines like triathlons
  10. The change in sex difference over years
    1. Pool-swimming
    2. Open-water long-distance swimming
    3. Swim split of longer triathlon races
  11. The influence of swimwear on performance
  12. Potential explanations for the female dominance
    1. Female dominance in long-distance swimming
    2. The influence of body fat on female swimming performance
    3. Other anthropometric and physiological characteristics of female swimming performance
    4. Psychological aspects and female swimming performance
    5. Kinematics and swimming performance
    6. Other aspects to be considered
  13. Conclusions

There are headers that concern types of swimming (pool, open water, ice, multi-sports disciplines), there are headers that discuss factors that influence performance (age, the rise of female sports over the years, performance level, swimwear) and finally, headers that discuss the explanations of the sex differences.

Answer: We reduced to Introduction, Method, Findings, Discussion, Conclusion.

Specific Comments

Line 135: It seems that the unit of measurement is incorrect. Either way, please be consistent throughout the manuscript.

Answer: The units are correct and taken from the published studies. However, for practical reasons, we express the times in minutes now in h:min. For the numbers expressed as swim speed and taken from the published studies, we changed from ‘performance’ to ‘swim speed’ to be consistent.

Line 426: space between “swimming” and “[59]”

Answer: We changed as suggested.

In conclusion…

Overall, this manuscript feels more like a book chapter or a piece in a popular scientific journal. Very interesting to read, a nice synthesis of a large body of information, but, it does not contribute much to the domain of science. Unless the authors manage to provide a bias-free coherent manuscript, I would not deem it fit for publication.

Answer: We changed the manuscript as suggested. However, it is not a book chapter but intended for future journalists to use for their newspaper articles as we mentioned in the ‘method section’. Journalists may profit for their future articles in magazines and newspapers since they have now all the important aspects summarized in one review paper.

Round 2

Reviewer 2 Report

The authors showed some improvement in their manuscript, and I would like to acknowledge their effort made in this short period of time. However, the manuscript still contains quite a lot of unnecessary information, and as I commented in the last review round, it still has many parts just listing the results where ‘a summary’ is rather required. The authors could make it much simpler and cut the number of references.

For example, I do not think it is appropriate to include details of a performance change or gender differences in each stroke, age, and calendar years reported in each article selected. As the authors do state in their manuscript, it is hugely affected by the individual sample differences and individual improvements of athletes competing each year, and there is no enough evidence suggesting general trends depending on different years and strokes. Therefore, I suggest the authors exclude such statements from every paragraph and just summarise what the references suggest (E.g., x, y, z% of selected studies suggested X, Y, Z, respectively. This inconsistency was probably due to…. However, there is a strong trend that female swimmers are more … than male swimmers). Of course, it is important to write some key numerical information. However, try to minimise it so that the flow of the text becomes much smoother than it is.

I must say a wide range of parts where the authors are supposed to discuss the reason why female swimmers could potentially outperform male swimmers (L450 onwards) are irrelevant. Despite the heading of the section (potential explanations for the female dominance), 80-90% of the section is just a general description of swimming performance determinants or some general anthropometric differences between males and females, and there are few sentences explaining the ‘why’. Please stick to the main message of the manuscript (which I understand as; Female swimmers tend to perform similar to or better than male swimmers in long-distance swimming and when they get older) and exclude all sentences and references that are not necessarily relevant.

Finally, I suggest the authors get their writing checked by a professional proofreader. There are quite a few unclear sentences (for example, use of comparison structure without stating comparing what and what in which variable) that would cause some misunderstandings for those who do not have much swimming knowledge. Furthermore, somehow the authors changed many sentences that explain general things to the past tense - for example, ‘In freestyle, 400 m, 800 m, 1500 m WERE additional races’ (L59-60). This example implies that it was the case in the past but not anymore (which is, obviously, not true). I do not know why the authors made such changes here and there, but this is not appropriate.

Author Response

Reviewer 2

The authors showed some improvement in their manuscript, and I would like to acknowledge their effort made in this short period of time. However, the manuscript still contains quite a lot of unnecessary information, and as I commented in the last review round, it still has many parts just listing the results where ‘a summary’ is rather required. The authors could make it much simpler and cut the number of references.

Answer: We agree with the expert reviewer and shortened the text with deleting references.

For example, I do not think it is appropriate to include details of a performance change or gender differences in each stroke, age, and calendar years reported in each article selected. As the authors do state in their manuscript, it is hugely affected by the individual sample differences and individual improvements of athletes competing each year, and there is no enough evidence suggesting general trends depending on different years and strokes. Therefore, I suggest the authors exclude such statements from every paragraph and just summarise what the references suggest (E.g., x, y, z% of selected studies suggested X, Y, Z, respectively. This inconsistency was probably due to…. However, there is a strong trend that female swimmers are more … than male swimmers). Of course, it is important to write some key numerical information. However, try to minimise it so that the flow of the text becomes much smoother than it is.

Answer: We agree with the expert reviewer and shortened and combined and deleted where possible.

I must say a wide range of parts where the authors are supposed to discuss the reason why female swimmers could potentially outperform male swimmers (L450 onwards) are irrelevant. Despite the heading of the section (potential explanations for the female dominance), 80-90% of the section is just a general description of swimming performance determinants or some general anthropometric differences between males and females, and there are few sentences explaining the ‘why’. Please stick to the main message of the manuscript (which I understand as; Female swimmers tend to perform similar to or better than male swimmers in long-distance swimming and when they get older) and exclude all sentences and references that are not necessarily relevant.

Answer: We agree with the expert reviewer and deleted where appropriate. However, some parts regarding age, motivation, etc. cannot be deleted regarding open-water ultra-distance swimming.

Finally, I suggest the authors get their writing checked by a professional proofreader. There are quite a few unclear sentences (for example, use of comparison structure without stating comparing what and what in which variable) that would cause some misunderstandings for those who do not have much swimming knowledge. Furthermore, somehow the authors changed many sentences that explain general things to the past tense - for example, ‘In freestyle, 400 m, 800 m, 1500 m WERE additional races’ (L59-60). This example implies that it was the case in the past but not anymore (which is, obviously, not true). I do not know why the authors made such changes here and there, but this is not appropriate.

Answer: We agree with the expert reviewer and changed to ‘In freestyle, the 800 m and the 1500 m are further race distances.’

Reviewer 3 Report

The authors have clearly responded to the raised concerns. Most of the issues have been resolved in the revised version of the manuscript. Particularly making the aim explicit to provide an overview for journalists makes it easier to put this narrative review into context.

This immediately brings me to my biggest remaining concern: I'm not sure if a scientific journal is the best venue for such a text. I still believe that this would be better suited as a book(chapter), where the aim to 'broadly inform' is clearer upfront.

That being said, I realize that I might be too narrow-minded in terms of what should and should not be published in a scientific journal. I'll leave it up to the editor to decide whether this kind of work serves the readership of the journal.

With that big concern out of the way, there are a few smaller issues that came to light in the second version of the manuscript.

1) be specific

In their writing, the authors sometimes make unspecific claims (that may or may not be specified later on in the text). The readability would improve if the authors as much as possible avoid unspecific claims, and replace them with claims that have the direction included.

An example:

Lines 71-72: "The author discussed the problem of comparing the fastest men and women in contrast to all men and women, leading to a different finding regarding male or female dominance [22].

Could be made more specific by stating that "the fastest men beat the fastest women, but that the average woman was faster than the average man" (which I copied from the actual newspaper article).

Other such statements are

Line 195-196: “Depending upon the sample size and the statistical approach...”

Please make it more specific.

2) separate journal and newspaper articles

In scientific literature, the default 'article' would be a scientific article. As this narrative review references newspaper articles, it is important to specify that difference when referring to newspaper articles.

For example:

Line 79: "In a further article" should be "In a further newspaper article".

Please make this explicit for all newspaper references.

3)

“Examining the relationship between sex and motivation in triathletes.”

I'm still not sure why this article hasn't been included. 

When I search for "sex difference swimming" (without quotation marks), I also find 558 results and on place 136 I find the article by "López-Fernández".

The paper is from 2014, so should have shown up, as far as I can tell.

Wouldn't it make the review appear more balanced if articles by other authors are included as well?

I would recommend to have another look at that.

4) minor spelling mistake

Some of the newly written text contains minor spelling mistakes:

Line 96: “Although the newspapers articles primarily” should be "newspaper articles"

Author Response

Reviewer 3

The authors have clearly responded to the raised concerns. Most of the issues have been resolved in the revised version of the manuscript. Particularly making the aim explicit to provide an overview for journalists makes it easier to put this narrative review into context.

This immediately brings me to my biggest remaining concern: I'm not sure if a scientific journal is the best venue for such a text. I still believe that this would be better suited as a book(chapter), where the aim to 'broadly inform' is clearer upfront.

That being said, I realize that I might be too narrow-minded in terms of what should and should not be published in a scientific journal. I'll leave it up to the editor to decide whether this kind of work serves the readership of the journal.

Answer: A journalist will not look for a book chapter when he is looking for a scientific article. All journalists who called me be phone or asked me by e-mail were looking for scientific studies published in journals. In order to make it easier for them we took the decision to write a review about this topic where all relevant studies are included.

With that big concern out of the way, there are a few smaller issues that came to light in the second version of the manuscript.

1) be specific

In their writing, the authors sometimes make unspecific claims (that may or may not be specified later on in the text). The readability would improve if the authors as much as possible avoid unspecific claims, and replace them with claims that have the direction included.

Answer: We agree with the expert reviewer and changed as suggested.

An example:

Lines 71-72: "The author discussed the problem of comparing the fastest men and women in contrast to all men and women, leading to a different finding regarding male or female dominance [22].

Could be made more specific by stating that "the fastest men beat the fastest women, but that the average woman was faster than the average man" (which I copied from the actual newspaper article).

Answer: We agree with the expert reviewer and changed as suggested.

Other such statements are

Line 195-196: “Depending upon the sample size and the statistical approach...”

Please make it more specific.

Answer: We agree with the expert reviewer and changed as suggested.

2) separate journal and newspaper articles

In scientific literature, the default 'article' would be a scientific article. As this narrative review references newspaper articles, it is important to specify that difference when referring to newspaper articles.

Answer: We agree with the expert reviewer and changed as suggested

For example:

Line 79: "In a further article" should be "In a further newspaper article". Please make this explicit for all newspaper references.

Answer: We agree with the expert reviewer and changed as suggested

3)

“Examining the relationship between sex and motivation in triathletes.”

I'm still not sure why this article hasn't been included. 

When I search for "sex difference swimming" (without quotation marks), I also find 558 results and on place 136 I find the article by "López-Fernández".

The paper is from 2014, so should have shown up, as far as I can tell.

Wouldn't it make the review appear more balanced if articles by other authors are included as well?

I would recommend to have another look at that.

Answer: We checked PUBMED again and found that article, but not with our key words. We inserted it as a reference in motivational aspects for female performance. 

4) minor spelling mistake

Some of the newly written text contains minor spelling mistakes:

Line 96: “Although the newspapers articles primarily” should be "newspaper articles"

Answer: We agree with the expert reviewer and changed as suggested.

This manuscript is a resubmission of an earlier submission. The following is a list of the peer review reports and author responses from that submission.